# Graph-based Reinforcement Learning meets Mixed Integer Programs: An application to 3D robot assembly discovery

**Niklas Funk**
Technical University of Darmstadt
niklas@robot-learning.de

**Svenja Menzenbach**
Technical University of Darmstadt
svenja.menzenbach@stud.tu-darmstadt.de

**Georgia Chalvatzaki**
Technical University of Darmstadt
georgia@robot-learning.de

**Jan Peters**
Technical University of Darmstadt
German Research Center for AI (DFKI)
Hessian.AI
Centre for Cognitive Science
jan@robot-learning.de

## Abstract

Robot assembly discovery is a challenging problem that lives at the intersection of resource allocation and motion planning. The goal is to combine a predefined set of objects to form something new while considering task execution with the robot-in-the-loop. In this work, we tackle the problem of building arbitrary, predefined target structures entirely from scratch using a set of Tetris-like building blocks and a robot. Our novel hierarchical approach aims at efficiently decomposing the overall task into three feasible levels that benefit mutually from each other. On the high level, we run a classical mixed-integer program for global optimization of blocktype selection and the blocks' final poses to recreate the desired shape. Its output is then exploited as a prior to efficiently guide the exploration of an underlying reinforcement learning (RL) policy handling decisions regarding structural stability and robotic feasibility. This RL policy draws its generalization properties from a flexible graph-based neural network that is learned through Q-learning and can be refined with search. Lastly, a grasp and motion planner transforms the desired assembly commands into robot joint movements. We demonstrate our proposed method's performance on a set of competitive simulated and real-world robot assembly discovery environments and report performance and robustness gains compared to an unstructured graph-based end-to-end approach. Videos are available at https://sites.google.com/view/milp-gnn-for-rad.

## 1 Introduction & Problem Definition

A common desire amongst many industry sectors is to increase resource efficiency. The construction industry could significantly reduce its environmental impact by re-using existing material more efficiently [1]. There is a fundamental need for combining intelligent algorithms for reasoning on how existing material can be recombined to form something new, with autonomous execution [2].

Herein, we tackle the problem of autonomous *robotic assembly discovery* (RAD), i.e., a robotic agent should reason about abstract 3D target shapes that

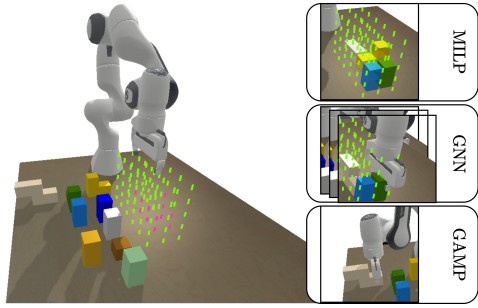

**Figure 1:** Illustrating a simulated RAD environment (left) and all three components of our proposed hierarchical approach (right).

N. Funk et al., Graph-based Reinforcement Learning meets Mixed Integer Programs: An application to 3D robot assembly discovery (Extended Abstract). Presented at the First Learning on Graphs Conference (LoG 2022), Virtual Event, December 9–12, 2022.

need to be fulfilled given a set of available building blocks (cf. Fig. 1). Unlike other assembly problems with known instructions, in RAD, the agent does neither have any prior information about which blocks to use and their final poses, nor about the execution sequence. Contrarily, the RAD agent should *discover* the possible ways of combining the building blocks, find appropriate action sequences, and put them into practice. RAD can thus be structured into two difficulty levels. On the high level, a goal-defined resource allocation problem has to be solved, which is typically NP-complete for discrete resources, and can be viewed as a real-world version of the Knapsack Problem [3]. The low level requires solving a motion planning problem, i.e., having to come up with an overall feasible action sequence of picking and placing actions taking into account the robot's kinematics, structural stability throughout the assembly, and avoiding any collisions.

One way of approaching RAD are end-to-end approaches that directly map from problem definition to low level actions [4–6]. They are typically straightforward to design, and based on learned graph neural network (GNN) representations. Due to their ability to learn relational encodings [7, 8] and invariant representations, they can overcome combinatorial barriers [9], and be combined with search for improved generalization [5, 6, 10]. Yet, they often require extensive training in combinatorial action spaces, and typically lack interpretability. On the other end of the spectrum are Task and Motion Planning approaches [11, 12], which naturally represent problem's hierarchy and necessitate full prior knowledge of geometrics and kinematics. They are usually unsuitable for real-time reactive control, as the full joint optimization suffers from combinatorics and non-convex constraints.

We propose a novel hierarchical method for 3D RAD that addresses both, resource allocation and motion planning. On the high level, a model-based mixed-integer linear program (MILP), handling the process of block-type selection and optimizing the blocks' final poses for optimally resembling the desired target shape, is solved. The MILP's solution is then used as a *guiding exploration* signal in a graph-based Reinforcement Learning (RL) framework. We define a GNN for capturing the geometric, structural, and physical relationships between building blocks, robot, and target shape, thereby incorporating all effects that have not been modelled on the higher level. The GNN is trained through model-free Q-learning allowing the integration with tree search for improved long-term decisions [10]. To put the previous reasoning into practice, at the lowest level, we rely on simple grasp and motion planning. We present an empirical evaluation of our proposed approach in a set of competitive simulated RAD tasks. The results show superior performance of our approach against both empirical and end-end GNN baselines, thereby underlining its effectiveness.

**Problem Definition**

We formulate the problem of having to combine rectlinear blocks into a desired target shape as Markov Decision Process. Its state is given by four sets: i) the set of unplaced blocks that encodes the remaining blocks, ii) the set of placed blocks that have already been used, iii) the set of target grid-cells (pink) that are part of the target shape and should all be filled, and iv) the set of non-target grid-cells (green) that should remain unoccupied (cf. Fig. 2). We also assume that all building blocks are a combination of primitive blocks. This choice allows to modularly represent any more complicated block through primitive elements.

For block placement, we use of a discrete, time-varying action space. Every unplaced primitive block can be placed w.r.t. all available grid-cells while additionally selecting from four actions that rotate the block by $0°$, $\pm 90°$, or $180°$ around the z-axis. We also add one termination action that results in stopping the assembly process. The resulting action space of combinatorial complexity thus contains #unplaced primitive elements$\times$#grid-cells$\times 4+1 = U_p \times G_c \times 4 + 1$ actions.

**Figure 2:** 2D RAD environment with one placed block consisting of two primitive elements (shown in brown/blue). The grid-cells are visualized through their centre points. Pink points represent target grid-cells that are to be filled & non-target grid-cells (green) should remain unoccupied.

After every placement action, the set of placed/unplaced elements and target/non-target grid cells are updated, and a reward is assigned. The reward is positive when the action reduced the number of target grid-cells, and negative if non-target grid-cells are being filled, therefore actively enforcing resource efficiency. The conditions for a successful placement action are that the block can be placed by the robot without moving or colliding with any other block, and that it is placed in a stable configuration. On any invalid action, the episode is terminated and a high negative reward is assigned.

Otherwise, the episode is terminated upon the events of i) the agent choosing the termination action, ii) no more available building blocks, or iii) the completion of RAD.

## 2  Method

We now introduce the two upper levels of our proposed tri-level hybrid approach for reliable RAD (cf. Fig. 1). For the lowest level that only realizes the commanded assembly actions, we refer to the appendix.

**High Level: MILP for optimal geometric target filling**. We first solve a MILP for optimizing the blocks' placing poses to optimally fill the desired shape in light of the problem's combinatorial complexity. Yet, to render the problem tractable, we do not consider the sequencing and robotic constraints. Based on the previous definitions (reward & voxelization), the MILP's objective (subject to maximization) equates to $\mathcal{O}_{\text{MILP}} = \boldsymbol{c}^T \boldsymbol{g}$, with vector $\boldsymbol{g} \in \mathbb{R}^{G_c \times 1}$ representing the grid-state, and $\boldsymbol{c} \in \mathbb{R}^{G_c \times 1}$ containing weights that indicate whether a grid-cell should be filled (1) or not (-1). We essentially flatten the 2D grid from Fig. 2 into a single vector by converting the discrete coordinates $d_x, d_y$ of every grid-cell to a single index $j = d_x + d_y n_x$ (with grid widht $n_x$). As every grid-cell can only be occupied at maximum by one block, we add $g[i] \leq 1, \forall g[i] \in \boldsymbol{g}$ as constraints. Next, we determine how every potential action changes the grid-state. I.e., placing the horizontal block from Fig. 2 in the lowest left position results in a grid state of $\boldsymbol{p}_{i=1, k=1}^T = [1, 1, 0, ...., 0] \in \mathbb{R}^{1 \times G_c}$, with block type index $i$ and placement action $k$. By additionally assigning a binary decision variable $w_{i,k}$ and taking all object types into account, we can define the change in the grid-state according to $\boldsymbol{g} = \sum_{\hat{i}=1}^{P} \sum_{\hat{k}=1}^{K(\hat{i})} w_{i=\hat{i}, k=\hat{k}} \boldsymbol{p}_{i=\hat{i}, k=\hat{k}}$ with a total of $P$ different block types and $K(i)$ admissible actions. While the binary decision variables prohibit any partial block placement by definition, we still have to restrict that any type of block can only be placed depending on its appearance in the scene $(N_i)$, $\sum_{\hat{k}=1}^{K(i)} w_{i, k=\hat{k}} \leq N_i, \forall i \in P$. We solve the resulting MILP through Gurobi [13] and obtain the optimal values for the decision variables thereby revealing the final poses for every block type.

**Medium Level: GNN for task sequencing**. The high level only partially resolves the problem's combinatorial aspect. It lacks i) the placement actions' sequencing, ii) the exact assignment of which block to use for each placement, and iii) the consideration of robotic feasibility, the blocks' initial positions, and structural stability. Thus, we require this level which is tasked to decide upon either executing one of the proposed actions from the higher level MILP or terminating the current assembly. We propose an approach based on combining GNN and Q-learning [5, 6]. The GNN is capable of providing the required representational flexibility and invariance to problem size, while performing action selection based on Q-learning is desirable as i) the action space is discrete, ii) the estimation of all the actions' quality as basis for action selection allows to efficiently incorporate the MILP's prior knowledge by masking out all actions that are not inside its solution, iii) potential multimodalities in the MILP solution can be captured, and iv) it allows easy and time-effective combination with search-based methods, i.e., Monte Carlo Tree Search (MCTS) to improve performance [10].
We now describe the action selection process (cf. Fig. 3). We refer to [6] (which esentially uses the same GNN) for the details. We first transform the environment's state into a graph by creating nodes for all primitive blocks and grid-cells. Every node has 5 initial features, i.e., the 3D world coordinates of its centre $\in \mathbb{R}^3$, and 2 booleans indicating the node type, i.e. placed/unplaced primitive block ($[1, 0]/[0, 0]$), target/non-target grid-cell ($[1, 1]/[0, 1]$). Almost all nodes of the graph are fully-

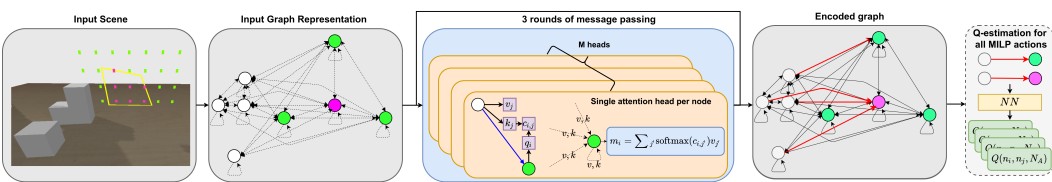

**Figure 3:** Illustrating action selection. First, the current scene is transformed into a graph. Note: Only a subset of the target (pink) and non-target (green) grid-cells is shown. White nodes depict the unplaced primitive blocks. Next follows message passing updating the nodes' features. The action's Q-values are predicted based on the nodes' features of the respective unplaced primitive block and the grid-cells using a feedforward neural network (NN). To incorporate the prior knowledge, we only consider actions part of the MILP solution (shown in red).

connected with each other – we only omit the connections in-between the unplaced primitive blocks if they do not belong to the same block to provide an inductive bias for the object shape. Upon graph creation follow three rounds of message passing using an attention mechanism [6, 9], in which we sequentially build an encoded graph. The encodings are the basis for computing Q-values for all available actions (i.e., predicting every actions' quality). As any unplaced primitive block can be placed w.r.t. every grid-cell, a standard feedforward NN is used, that takes as input the encoded node values of i) the primitive block-to-be-placed, and ii) the grid-cell, and outputs the Q-values for all the four rotational-placement actions between these nodes. This process is repeated for all pairs of unplaced primitive blocks and grid-cells. The action decision is done using an $\epsilon$-greedy policy, yet, only allowing to choose from the set of actions proposed by the MILP (we mask out all the other ones), as well as the termination action. The $\epsilon$-greedy policy controls the tradeoff between randomly exploring actions and exploiting, i.e., selecting the action with highest Q-value during training and evaluation. The graph's weights are refined through temporal-difference learning, thereby attempting to improve the estimation of the Q-values by minimizing the difference between the predicted quality of the action and the observed outcome. While this Q-learning procedure by itself already results in good policies, during test time, we additionally consider action selection based on the combination of Q-learning and MCTS (DQN+MCTS). For more details, please see [14] and the appendix.

## 3    Experimental Results

We evaluate our proposed MILP-DQN method and potentially adding MCTS (search budget of 5), in simulation (Fig. 1) and reality (cf. link to videos in abstract). We aim to answer two questions: 1) Does the MILP's guided exploration signal improve performance compared to end-to-end approaches? 2) How effective is the medium level GNN compared to an heuristic approach for task sequencing? The training is conducted as in [6], and we also reuse their simulation, yet, allowing block placements throughout the whole assembly area & voxelizing the target shape. In the evaluations, we describe the environment's difficulty through the grid size, i.e., Fig. 2 shows a potential target shape for a grid size of 3. The star(*) indicates the agents' evaluation in their training conditions, while the other experiments are out-of-distribution. The results are obtained by averaging the agents' performance in 200 scenes. We report the discounted reward $R$, the fraction of runs that ended upon failure $f$, and the target grid-cell coverage $\bar{a}$, i.e., the fraction of initially unfilled and finally filled target grid-cells.

*A) Is the high level MILP needed?*

We consider scenarios without the robot, which reduces the task's complexity to placing the blocks in a stable configuration while trying to optimally fill the desired shape. We compare against two baselines. The first one (DQN) does not consider the MILP's prior knowledge and can therefore place any of the available blocks at all currently unoccupied grid-cells. The second one (DQN-REL) follows [6], in which the available blocks can only be placed next already placed blocks, thus, reducing the action space. In the first step, we allow to place the blocks at any target grid-cell.

**Table 1:** Comparing our proposed method with two learned baselines in the two-sided environment wo robot.

| Grid Size | Method | $R$ | $\bar{a}$ |
|---|---|---|---|
| 3* | DQN | 0.63 (0.02) | 0.71 |
|  | DQN-REL [6] | 0.67 (0.01) | 0.68 |
|  | **MILP-DQN** | **1.22** (0.01) | **0.87** |
| 4 | DQN | 0.71 (0.08) | 0.69 |
|  | DQN-REL [6] | 0.75 (0.08) | 0.66 |
|  | **MILP-DQN** | **1.56** (0.03) | **0.87** |
| 5 | **MILP-DQN** | **1.92** (0.05) | **0.85** |

The results in Table 1 reveal that the MILP provides a strong inductive bias that is effective in guiding the exploration. The agents trained using our proposed MILP-DQN approach outperform the two baselines which in turn exhibit very similar performance. Compared to the baselines, MILP-DQN agents achieve an increase in the success rate and discounted reward by a factor of 2. These results confirm the task's combinatorial complexity. Performing an $\epsilon$-greedy exploration without using an informed prior does not allow for discovering good action sequences. The results also reveal that the MILP-DQN agents generalize well to the out-of-distribution environments as the desired target grid-cell coverage remains high at 0.87 and 0.85 (grid size of 4,5), despite the significant increase in task complexity. I.e., the number of blocks in the scene increases in line with the average target grid-cells that should be filled. The latter increases from roughly 5 to 12 while increasing the grid size from 3 to 5.

*B) How effective is the GNN for robotic execution?* We now consider the scenario with the robot (Fig. 1) and investigate the GNN's effectiveness. For this purpose, we compare the GNN with a heuristic (HEUR). The HEUR agents perform action selection as follows: based on MILP's proposed actions,

the heuristic only considers those which will result in a stable block placement and selects one of them at random. If there is no such action, the termination action is selected.

As shown in Table 2, in both environments, our proposed agents (MILP-DQN & MILP-DQN-MCTS) clearly outperform the heuristic. Already in the environment with less blocks, the heuristic results in 40% of failures, indicating that a more informed method for action sequencing is required. An example of such a failure is depicted in Fig. 5, where due to bad action sequencing by the heuristic, the two blocks collide. Our proposed approaches effectively reduce the failure rates, with MILP-DQN achieving a decrease by a factor of 2, while adding MCTS leads to

**Table 2:** Comparing our proposed method with a heuristic in the two-sided environment with the robot-in-the-loop.

| Grid Size | Method | $R$ | $f$ | $\bar{a}$ |
|---|---|---|---|---|
| 4* | HEUR | 0.57 | 0.4 | 0.62 |
| | **MILP-DQN** | 1.03 | 0.16 | 0.7 |
| | **MILP-DQN-MCTS** | **1.24** | **0.05** | **0.75** |
| 5 | HEUR | 0.34 | 0.58 | 0.47 |
| | **MILP-DQN** | 0.98 | 0.25 | 0.58 |
| | **MILP-DQN-MCTS** | **1.38** | **0.08** | **0.65** |

a decrease by a factor of almost 8. Those results show that our learned graph-based representations are indeed capable of effectively capturing the environment's state and make informed decisions regarding the action sequencing - a crucial component of RAD. Overall, we conclude that our proposed hierarchical approach is indeed capable of resolving the inherent difficulties of RAD, as also illustrated in Fig. 4 where we show the successful assembly of a desired target shape using 4 blocks of 3 different types. Moreover, on our website, we also showcase real-world transfer of the learned policies.

## 4 Conclusions

We have presented a novel hierarchical approach for robot assembly discovery (RAD). Our approach combines global reasoning through mixed-integer programming, which forms a powerful inductive bias for the subsequent graph-based reinforcement learning for local decision-making, together with grasp and motion planning for realizing the assembly actions. The hierarchy efficiently decomposes the problem's huge combinatorial action space and results in robust and reliable RAD policies. The proposed approach is validated in a set of simulated RAD and real-world experiments that illustrate its effectiveness. As graph structures are already widely used in robotics (i.e., kinematic/dynamic chains, scene graphs, factor graphs), in the future, we want to investigate how our approach and learning on graphs can be applied in different problem settings and domains.

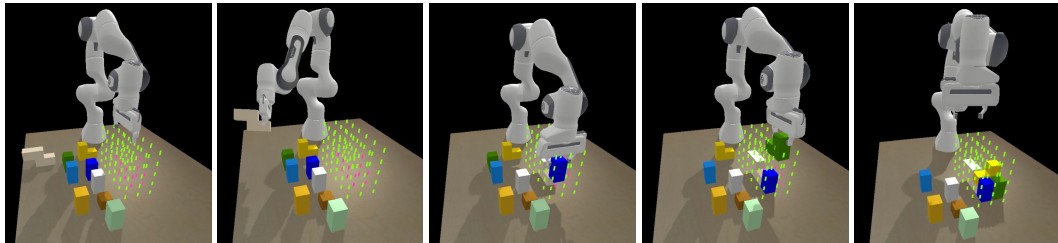

**Figure 4:** Illustration of a successful RAD sequence using our proposed MILP-DQN-MCTS approach. The agent successfully the assembly successfully using in total 4 blocks and 3 different block types.

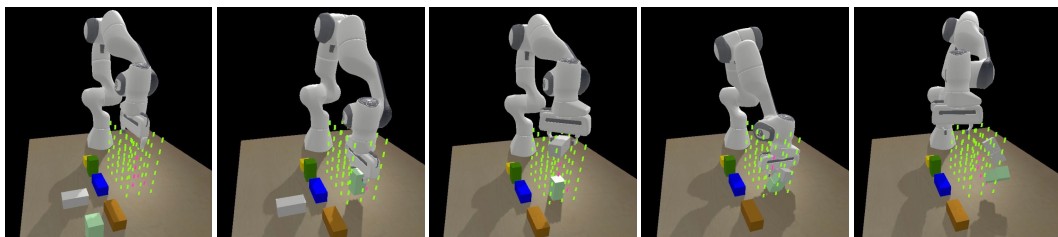

**Figure 5:** Illustration of an unsuccessful RAD sequence using the heuristic agent introduced in Sec. 3-B. As shown in the images, it is important to perform informed decisions about the assembly sequence, as the wrong sequencing can result in collisions between the block that is placed and other blocks in the scene.

## Acknowledgements

This work is supported by the AICO grant by the Nexplore/Hochtief Collaboration with TU Darmstadt, and the Emmy Noether DFG Programme (No. 448644653). Calculations for this research were conducted on the Lichtenberg high performance computer of the TU Darmstadt.

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

# A  Appendix

## A.1  Visualization of successful RAD sequences

To support the experimental evaluations presented in Section III-B, we also present videos on our website https://sites.google.com/view/milp-gnn-for-rad that illustrate the difference between the agents.

We also show on the website that we can successfully transfer our learned policies to real-world RAD instances. To get all the information about the initial scene, we use an OptiTrack motion capture system. Afterwards, we can use this information to create a digital twin and subsequently plan in simulation and execute the respective actions also in the real world as shown in the videos.

## A.2  Further details regarding our proposed approach

In this section, we aim to summarize the overall working of our algorithms and provide a more thorough description of the individual components that are involved.

### A.2.1  Formulating RAD as MDP

As already mentioned in the problem definition section, we describe RAD using the notation of a Markov Decision Process (MDP) with state and action space, $\mathcal{S}, \mathcal{A}$, transition probabilities $p$, reward function $r$, and discount factor $\gamma$.

**State space**

The state $s$ is given by the combination of four sets, $s=(\mathcal{S}_U, \mathcal{S}_P, \mathcal{T}_F, \mathcal{T}_E)$, with $|\mathcal{S}_U|=N_U, |\mathcal{S}_P|=N_P, |\mathcal{T}_F|=N_F, |\mathcal{T}_E|=N_E$. The set $\mathcal{S}_U$ encodes the unplaced primitive units that are still available for construction, $\mathcal{S}_P$ the primitive units that have already been used, $\mathcal{T}_F$ and $\mathcal{T}_E$ contain the so-called target grid-cells and non-target grid-cells, respectively. These grid-cells are parameterized through their respective 3D center coordinate $\boldsymbol{x} \in \mathbb{R}^3$, i.e. $\mathcal{T}_F=\{\boldsymbol{x}_i, i \in N_F\}$, $\mathcal{T}_E=\{\boldsymbol{x}_i, i \in N_E\}$ (visualized in pink and green). By projecting all grid-cells centre coordinate $\boldsymbol{x}_i$ into the yellow target shape (cf. Fig. 2), we decide whether it should be occupied or remain unoccupied during RAD.

Moreover, we assume that all building blocks are a combination of primitive units. More specifically, we consider that there is only one type of primitive unit: a unit cube $u_c = 1^3$. Thus, all the blocks in the scene are a combination of primitive units (cf. Fig. 2), i.e., block $i$ is defined by the union of $N_{b_i}$ primitive units, $b_i = \bigcup_{j=1}^{N_{b_i}} u_c$. Representing blocks as concatenations of primitives allows for a universal interface with graph-based representations, as any Tetris-like block can easily be represented. Simply put, each primitive unit induces a node in the graph, and the connectivity information encodes whether or not multiple primitive units form a larger block (cf. two leftmost frames in Fig. 3). This choice also allows us to describe the placed and unplaced blocks through the primitive units' 3D positions $\boldsymbol{x}_k$, and connectivity information $y_k = [y_{k,1}, ..., y_{k,N_U}]$, i.e., $\mathcal{S}_U=\{(\boldsymbol{x}_k, y_k), k \in N_U\}$. If primitive unit k is connected with primitive unit 1 to form a larger block $y_{k,1}$ equals 1, otherwise $y_{k,1}=0$. We follow the same procedure for the set of already placed elements $\mathcal{S}_P$.

**Action space**

For placing blocks in the scene, we use a discrete, time-varying action space. In particular, every primitive unit which is at the moment unplaced, can be placed w.r.t. all available grid-cells. As more complicated blocks might also require rotations, we augment all placement actions with four rotational actions, i.e. rotating the block by $0, \pm90$, or 180 degrees around the upward-pointing z-axis. Furthermore, we add one termination action that enables the agent to indicate that the current assembly is finished or not possible to continue, as there are no feasible actions left. Thus, the resulting action space contains $N_a = N_U \times (N_F + N_E) \times 4 + 1 = N_U \times G_c \times 4 + 1$ actions. Note that the MDP is focused on high level decision making. It does not account for the low level motion generation, namely grasp selection and robot motion planning, as this would further increase the already large action space. Nevertheless, given the action, the motion generation problem is well defined as it specifies the block that is to be moved, the required relative change in orientation, and its placement location. After every placement action, all primitive units belonging to the moved block are transferred from the set of unplaced elements to the set of placed ones. We also update the set of grid-cells by removing all cells that are now occupied.

**Reward definition**

On every successful placement action, we assign a reward of $r(s_t, a_t) = 0.2(N_{F_t} - N_{F_{t+1}} + N_{E_{t+1}} - N_{E_t})$, thereby giving a positive signal when the action reduced the number of target grid-cells, while also penalizing unnecessary filling of non-target grid-cells. Thus, this choice actively enforces resource efficiency. The conditions for a successful placement action (i.e., valid action) are:

- the block is placed by the robot without moving or colliding with any other block

- the block is placed in a stable configuration (i.e. the resulting structure is not falling apart due to gravity).

If any of these conditions is violated, the action will be marked as invalid. This results in terminating the current episode and assigning a reward of $-1$.

To summarize the previous definitions, the reward is given by

$$r(s_t, a_t) = \begin{cases} 0.2(N_{F_t} - N_{F_{t+1}} + N_{E_{t+1}} - N_{E_t}) & \text{if valid action executed,} \\ -1 & \text{if executing an invalid action,} \\ 0.2(N_{F_t} - N_{F_{t+1}} + N_{E_{t+1}} - N_{E_t}) + 1 & \text{if valid action and } N_{F_{t+1}} == 0, \text{ i.e., the completion of RAD,} \end{cases}$$
$$(1)$$

As the last case corresponds to the desired behaviour, i.e., successful completion of RAD, we increase the final reward by $+1$ upon this event.

**Episode termination**

We additionally want to point out which events result in terminating the current episode. Terminating the episode requires having to sample a new RAD scene before taking the next action. Every episode terminates upon one of the following events occurring:

- the agent selecting an invalid action (i.e., one of the following: 1) upon block placement, the robot or the block, or both collide with any other block, 2) the block placement results in an unstable configuration, i.e., the resulting structure falling apart due to gravity)

- the agent choosing the termination action

- no more available building blocks

- the completion of RAD, i.e., the filling of all target grid-cells.

**Discount factor**

Finally, to reflect the long-horizon of the considered task, we set the discount factor $\gamma$ to 0.999. For the definition of the reward, we refer to the next section.

### A.2.2    High Level: MILP for optimal geometric target filling

The pseudocode in Alg. 1 contains all the logic to obtain and update the MILP's solution. In particular, the function *computeMILPSol* from Line 4 onwards describes the necessary steps to obtain the MILP solution given the current state. The pseudocode closely follows our descriptions from Sec. 2 for the highest level.

As the environment state changes constantly during the assembly process, we also require another function that updates the actions that are available to the agent. This function is called *updateAvailActions* and described from Line 16 onwards. Please note that the update function does not require solving the MILP again (it takes as input the previously calculated solution) and is therefore way more efficient.

## Algorithm 1 MILP

1: Grid is of size $n_x, n_y, n_z$ (x-, y-, z-direction, respectively)
2: Grid has $G_c = n_x n_y n_z$ cells
3: Conversion from grid coordinate $d_x, d_y, d_z$ to index via $j = d_x + d_y n_x + d_z n_x n_y$
4: **procedure** $computeMILPSol(s)$
5:      Extract grid state $\boldsymbol{g} \in \mathbb{R}^{G_c \times 1}$ from $s$
6:      Add $G_c$ constraints ensuring only 1 primitive block at each cell: $g[i] \leq 1, \forall g[i] \in \boldsymbol{g}$
7:      From $\mathcal{S}_U$, extract the $P$ different block types that are in the scene, and their quantities $N_i$
8:      $K, \mathbf{M} = computeUniquePlacements(\mathcal{S}_U, P, \boldsymbol{g})$ (cf. Line 20)
9:      Introduce vector of binary decision variables: $\boldsymbol{w} \in \{0, 1\}^{(\sum_{i=1}^{P} K(\hat{i})) \times 1}$
10:     Add contraint on decision variables regarding occurance of each block type: $\sum_{l=K(i-1)}^{K(i)} w[l] \leq N_i$, for i in 1..P, and with $K(0) = 1$
11:     Define cost vector $\boldsymbol{c} \in \mathbb{R}^{G_c \times 1}$
12:     Define Objective: $\max_{\boldsymbol{w}} \boldsymbol{c}^T \mathbf{M} \boldsymbol{w}$
13:     Solve optimization problem, returns optimal entries for $\boldsymbol{w}$
14:     compute all available actions: $\mathcal{A}_{\text{MILP}} = computeAvailGNNActions(\mathcal{S}_U, \boldsymbol{g}, \boldsymbol{w}, \mathbf{M})$ (cf. Line 34)
15:     **return** $\mathcal{A}_{\text{MILP}}, \boldsymbol{w}, \mathbf{M}$
16: **procedure** $updateAvailActions(s, \boldsymbol{w}, \mathbf{M})$
17:     Extract grid state $\boldsymbol{g} \in \mathbb{R}^{G_c \times 1}$ from $s$
18:     compute all available actions: $\mathcal{A}_{\text{MILP}} = computeAvailGNNActions(\mathcal{S}_U, \boldsymbol{g}, \boldsymbol{w}, \mathbf{M})$
19:     **return** $\mathcal{A}_{\text{MILP}}, \boldsymbol{w}, \mathbf{M}$
20: **procedure** $computeUniquePlacements(\mathcal{S}_U, P, \boldsymbol{g})$
21:     Define empty Matrix $\mathbf{M}$
22:     **for** $\hat{i}$ in 1..P **do**
23:         Number of valid placing actions: $K(\hat{i}) = 0$
24:         **for** $m$ in 1..$G_c$ **do**
25: /* Iterate over all potential placing positions
26:             **for** $h$ in 1..4 **do**
27: /* Iterate over all rotational actions
28:                 Starting from empty grid with all zeros: $\boldsymbol{g}_1 = 0^{G_c \times 1}$
29:                 Attempt to place the first primitive unit $p_1$ of block $\hat{i}$ at grid-cell $m$ while applying rotation $h$, and capture resulting grid-state $\boldsymbol{g}_1$
30:                 **if** All primitive units inside cell && $\boldsymbol{g}_1$ not equal to any column in $\mathbf{M}$ && $\boldsymbol{g}_1^T \boldsymbol{g} = 0$ **then**
31:                     Append vector $\boldsymbol{g}_1$ as a new column to $\mathbf{M}$
32:                     $K(\hat{i}) += 1$
33:     **return** $K, \mathbf{M}$
34: **procedure** $computeAvailGNNActions(\mathcal{S}_U, \boldsymbol{g}, \boldsymbol{w}, \mathbf{M})$
35:     Define empty list of available actions $\mathcal{A}_{\text{MILP}}$
36:     From $\mathcal{S}_U$, extract the $P_t$ different block types that are currently in the scene
37:     **for** $\hat{i}$ in 1..$P_t$ **do**
38:         **for** $m$ in 1..$G_c$ **do**
39: /* Iterate over all potential placing positions
40:             **for** $h$ in 1..4 **do**
41: /* Iterate over all rotational actions
42:                 Starting from empty grid with all zeros: $\boldsymbol{g}_1 = 0^{G_c \times 1}$
43:                 Attempt to place the first primitive unit $p_1$ of block $\hat{i}$ at grid-cell $m$ while applying rotation $h$, and capture resulting grid-state $\boldsymbol{g}_1$
44:                 **if** All primitive units inside cell && $\boldsymbol{g}_1$ equal to any column $j$ in $\mathbf{M}$ for which holds $\boldsymbol{w}[j] == 1$ && $\boldsymbol{g}_1^T \boldsymbol{g} = 0$ **then**
45: /* This if checks: 1) is the block entirely in the target area?, 2) is the action that is to be executed inside the solution space?, 3) is any of the voxels, where the block might be placed already filled?
46:                     Append Triple $(p_1, m, h)$ to $\mathcal{A}_{\text{MILP}}$
47:     Append termination action $(0, 0, 5)$ to $\mathcal{A}_{\text{MILP}}$
48:     **return** $\mathcal{A}_{\text{MILP}}$

### A.2.3   Medium Level: GNN & Q-Learning for task sequencing.

On the medium level, we now get as input the solution from the higher level and use it to train our graph-based reinforcement learning agents.

To start with, we present the general loop that is used to train the graph-based representations in Alg. 2. It consists of two main components. We first collect experience through interacting with the environment (cf. while loop in Line 6), and secondly, we use the obtained samples to refine our GNN (cf. Line 23) that estimates the quality of all the actions and thus directly influences the actions that are being taken in the environment.

---

**Algorithm 2** Training Loop for the medium level GNN-RL

---

1: **for** $i$ in 1..NumberEpochs **do**
2: /* Collect experience
3:     $j = 0$, Buffer $\mathcal{B} = []$
4: /* Define number of samples to collect
5:     $\Gamma = 100$
6:     **while** $j < \Gamma$ **do**
7:         Sample initial state $s$
8: /* Obtain MILP solution by running *computeMILPSol* from Alg. 1, cf. Line 4
9:         $\mathcal{A}_{\text{MILP}}, \boldsymbol{w}, \mathbf{M} = computeMILPSol(s)$
10:         finished=False
11:         **while** finished==False **do**
12:             Sample action $a = act(Q, s, \mathcal{A}_{\text{MILP}})$ using Q-function approximator $Q$. This calls Alg. 3 Line 2
13:             Execute $a$, i.e., move robot to pick and place the part & obtain $r(s, a)$
14:             Receive next state $s'$
15:             $\mathcal{B}$.append($[s, a, r(s, a), s']$)
16:             $j = j + 1$
17:             $s = s'$
18: /* Update MILP solution by running *updateAvailActions* from Alg. 1, cf. Line 16
19:             $\mathcal{A}_{\text{MILP}} = updateAvailActions(s, \boldsymbol{w}, \mathbf{M})$
20:             **if** Any of the termination criteria (cf. A.2.1) is true **then**
21:                 finished=True
22: /* Update weights of Q-function
23:     $\pi = update(Q, \mathcal{B})$

---

During training and also during evaluation of our proposed MILP-DQN approach, we perform action selection as shown in Alg. 3 Line 2. In both cases of either exploring a random action (cf. Line 5) or selecting the action with highest predicted Q-value (i.e., exploitation, cf. Line 7), we only allow to choose from the set of actions that has been previously proposed by the high level MILP. For training the GNN to predict the correct Q-values, we exploit the collected experience and perform temporal difference learning, as shown in Alg. 3 from Line 8 onwards.

---

**Algorithm 3** DQN

---

1: Number of update steps $\chi$
2: **procedure** $act(Q, s, \mathcal{A}_{\text{MILP}})$
3: /* $\epsilon$-greedy policy
4:     **if** $\text{RandomVariable} < \epsilon$ **then**
5:         $a = \text{RandomChoice}(\mathcal{A}_{\text{MILP}})$
6:     **else**
7:         $a = \max_{a'} Q(s, a'|a' \in \mathcal{A}_{\text{MILP}})$
        **return** $a$
8: **procedure** $update(\pi, \mathcal{B})$
9:     Add $\mathcal{B}$ to Replay Memory
10:     **for** $i$ in 1..$\chi$ **do**
11:         Sample random subset from Replay Memory
12: /* Temporal-difference learning using tatget network $Q_T$ as in [15].
13:         $\text{loss} = \text{smoothL1}(Q(s, a) - (r(s, a) + \gamma \max_{a'} Q_T(s', a'|a' \in \text{AllPossibleActions}(s))))$
14:         Update Q-function approximator $Q$ with parameters $\theta$
15:         $\theta = \theta - \alpha \frac{\partial \text{loss}}{\partial \theta}$
16:     **return** $Q$

---

Lastly, we are only missing the details regarding action selection for our proposed MILP-DQN-MCTS method. Contrarily to the previous MILP-DQN approach, here, we even add model-based search through MCTS. This has the potential to even further improve performance, robustness, and generalization.

Alg. 4 provides the details for action selection in MILP-DQN-MCTS agents. Please note, that Alg. 4 is still only capable of selecting from the set of actions proposed by the high level MILP. As can be seen in the pseudocode, we now simulate the outcome of multiple actions and subsequently exploit this experience to decide upon the desired action that should be executed. We provide the code for the search process from Line 7 onwards. Moreover, as pointed out in the second line of the algorithm, we only consider a rollout depth of 1. This means that we stop the model-based rollouts after the first action and estimate the expected reward of the remaining trajectory by again querying our Q-function estimator. This possibility of clipping the rollouts already after the first or generally speaking after very few actions is another reason why the combination of Q-learning and MCTS is appealing and efficient.

---

**Algorithm 4** DQN + MCTS

---

1: /* Note, this is only during evaluation.
2: Rollout Depth $\eta = 1$ if not stated otherwise
3: Search Budget $\tau = 5$ if not stated otherwise
4: **procedure** $act(Q, s, \mathcal{A}_{\text{MILP}})$
5:     Given: state $s$, set containing the explored actions $\mathcal{S}_A = \{\}$
6:     $\forall a \in \mathcal{A}_{\text{MILP}}$, Initialize $W(s, a) = 1$, $Q_S(s, a) = Q(s, a)$
7:     **for** $i$ in $1..\tau$ **do**
8:         **if** RandomVariable $< \epsilon$ **then**
9:             $a = \text{RandomChoice}(a \in \mathcal{A}_{\text{MILP}} | W(s, a) = 1)$
10:         **else**
11:             $a = \max_{a'} Q(s, a' | a' \in \mathcal{A}_{\text{MILP}}, W(s, a') = 1)$
12:         Add $a$ to $\mathcal{S}_A$, collect $r(s, a)$, update $\mathcal{A}_{\text{MILP}} = $ *updateAvailActions*$(s, \boldsymbol{w}, \mathbf{M})$
13:         **for** $j$ in $1..\eta - 1$ **do**
14:             $a = \text{DQN} - act(Q, s)$ (Alg. 3, Line 2), collect current single step reward $\tilde{r}$
15:             Update: $r(s, a) = r(s, a) + \gamma^j \tilde{r}$ and $\mathcal{A}_{\text{MILP}}$
16:         Update: $W(s, a) = W(s, a) + 1$, $Q_S(s, a) = \frac{1}{2}(Q_S(s, a) + r(s, a) + \gamma^\eta \max_{a'} Q(s', a'))$
17:     $a_r = \max_{a'} Q_S(s, a' | a' \in \mathcal{S}_A)$
18:     **return** $a_r$

---

### A.3 Additional details on the lowest level: Grasp and Motion planning (GAMP)

The lowest level is tasked with the conversion of the previous level's actions into robot joint commands, and performs the final robot execution of block grasping and moving such that the block is placed in the desired pose. While it would be possible to add those decisions to the higher levels, we decided to consider motion generation as a separate module in our hierarchical framework, as these decisions are heavily dependent on the actual robot manipulator. Moreover, we want to avoid increasing the action space of the previous level. Robotic block grasping and placing is achieved by first checking the feasibility of a predefined set of top-down grasping poses and subsequently checking if this grasp results in a feasible final placement pose. If there exists a pair of feasible grasping and placing poses, we move the robot by approaching the grasping pose from the top, then move to a position that is slightly above the placing location, and finally, approach the placement pose. All intermediate waypoints are computed based on inverse kinematics.

### A.4 Additional details on running times

Lastly, we want to provide the running times of our individual components. We focus on the environment with the robot-in-the-loop, and thus report the running times for the experiments presented in Section 3 - B). Please note that we did not have time to properly optimize our code, thus, we think that there is still lots of room for improvement for the running times that we will report in this section. The results are again obtained by averaging across all the 200 RAD scenes that have been presented to the agents. We also want to emphasize that computing the initial MILP solution is only required once per scene, whereas all the other components have to be run per action, i.e., per step that is taken in the environment.

The results from this experiment are shown in Table 3. Computing the initial MILP solution, i.e., calling the function *computeMILPSol* (cf. Alg. 1), takes around 18 milliseconds (ms), and 26 ms for the environments with a grid sizes of 4 and 5, respectively. Please again note, that computing the

**Table 3:** Reporting the average running times of all our components in the same experimental setting as presented in Section 3 - B). All the running times have been acquired on a computer with 64GB RAM, an NVIDIA GeForce RTX 2080 SUPER GPU, as well with an AMD Ryzen 9 3900x CPU (24 cores).

| Grid Size | Compute MILP solution (**per scene**) | Update MILP solution (per step) | GNN-DQN (per step) | GNN-DQN + MCTS (per step) | GAMP (per step) |
|---|---|---|---|---|---|
| 4 | 0.0178s | $6.5410^{-5}$s | 0.0069s | 1.2094s | 0.0324s |
| 5 | 0.0259s | $7.5110^{-5}$s | 0.0069s | 1.3968s | 0.0370s |

initial MILP solution is only required once for every RAD scene. All the other components have to be run for every action, i.e., every step that is taken in the scene. Further, the table shows that updating the MILP solution, i.e., calling the function *updateAvailActions* (cf. Alg. 1) requires by far the least amount of time and is negligible compared to the other running times. Calculating the desired action based on the GNN-DQN approach only (cf. Alg. 3) is also very efficient as it only takes about 7ms for both of the environments. However, if we take more than 3 actions per RAD scene, then the total time required for the GNN-DQN already exceeds the time taken to compute the initial MILP solution. Moreover, running the grasp and motion planning (GAMP, cf. Sec. A.3) which is required for every action requires on average around 32 and 37ms (for the two different versions of the environment) and thus even consumes more time than computing the initial MILP solution. Finally, when performing the action decision based on the combination of GNN-DQN + search(MCTS) as described in Alg. 4, this requires 1.2s and even 1.4s on average for the environments with the grid sizes of 4 and 5. The big increase in runtime compared to the GNN-DQN approach can be explained by the fact that we explore five different actions before we decide upon the one that should be taken. This means that we have to query the GNN five times, perform GAMP five times, and lastly, have to evaluate the outcomes of the five actions using our PyBullet simulation which is very costly. Nevertheless, we still want to point out that our approach is targeted at high-level decision-making and that the robot motion in the real world (i.e., picking and placing the block) takes on the order of 20s, which is still much longer than the time taken to decide upon the action. However, as we plan to apply our proposed algorithms to different domains, speeding up this combination of DQN+MCTS is on top of our priority list as it performed best across experiments.

