# OpenReview forum: "Graph-based Reinforcement Learning meets Mixed Integer Programs: An application to 3D robot assembly discovery"
_logconference.io/LOG/2022/Conference — LoG 2022 Poster_

### Official Review · Reviewer_E4Cb · 2022-10-17

**Overall Score:** 6
**Confidence:** 2

**Review:**

Summary:

This paper proposes to solve robot assembly discovery using a combination of mixed integer programming and GNN-based reinforcement learning. The main contribution of this paper is from constraining the RL action space using prior knowledge about the environment. Namely, the main idea is to constrain the GNN-based agent to take only the actions that are included in the integer programming solution. Such a GNN agent is needed since the integer programming solution only decides the set of block locations, but does not decide which order should the block be placed.

Strenth & Weakness:

S1 This paper proposes a reasonable idea that one can divide the problem into an integer-programming-solvable part and an RL part.

W1 The contribution of this paper (combining integer programming with RL) is not very relevant to LOG conference. This paper utilizes GNN as the backbone for its RL agent, but the GNN was proposed in prior work.

W2 The methodological contribution of this paper (constraining solution space using prior knowledge) is not very novel in the ML/graph community. For example, S2V-DQN [a] constrains the solution space using prior knowledge about the combinatorial optimization problem being solved. The only baseline (DQN-REL [6]) constrain the decision to place blocks next to already placed blocks

W3 Some details could be more clarified. How exactly does the GNN agent incorporate the MILP solution? Is the experimental environment newly developed in this paper, or taken from [6]?

W4 Empirical evaluation could be stronger. If the environment is taken from [6], why not compare it with the same baselines and varying numbers of blocks? Reporting the standard deviation would also be useful since the number of experiments is small (may be fragile to high variance).

W5 Why only use DQN-MILP for Grid size 5 in Table 1?

W6 Why not compare with DQN and DQN-REL for Table 2?

Recommendation:
I recommend "weak reject" for this paper. My main concerns are (1) lack of contribution & novelty with respect to the LOG conference and (2) experimental results that are not very convincing.

[a] Learning Combinatorial Optimization Algorithms over Graphs, NeurIPS 2017

---

### Official Review · Reviewer_ezuy · 2022-10-20

**Overall Score:** 6
**Confidence:** 5

**Review:**

# Summary of The Paper
The paper is about the application of robotic assembly discovery, that
is to place several predefined primitive blocks into a target 3D
shape. The proposed method for solving this problem is a (novel to my
knowledge) combination of three known elements: (1) a mixed integer
programming formulation, which is to determine whether a grid cell should be
filled or not; (2) a graph neural network combined with deep Q
learning and Monte Carlo tree search, which is to decide the sequence
of actions that should be taken for the placements of the primitive
blocks; (3) a grasp and motion planning module, which is to turn the
abstract actions into actual commands for the robot to execute.

# Strength
1. The paper is well written. One could feel the efforts of the
   authors in trying to avoid mathematical formulas and aiming to
   describe their approaches using natural language in a smooth way.
2. Blending GNNs and MILPs into a working algorithm for a
   practical application is very interesting.
3. The two experiments are well-designed and are great ablation
   studies. The first experiment in Table 1 illustrates that solving
   a MILP could yield a graph representation that is able to better
   guide the exploration of the GNNs + deep Q learning. Table 2 shows
   that the reinforcement learning approach, which learns the action
   sequence as heuristics, is better than the heuristic designed by
   simple rules (HEUR).

# Weakness and Questions
1. The novelty is a bit limited. It is a basic combination of existing
   approaches.

2. Some other experiments are desired. For example, (1) would the
   proposed approach run slower than using MILPs or GNNs alone? The
   authors are encouraged also to report the running times; (2) The
   paper considers a real problem, but the experiments seemed to be
   conducted in some simulation environments. It would be nice and
   make the paper more convincing if some real experiments are
   presented. In particular, a demo video in a real-world setting
   would better illustrate the power of the proposed idea.

3. The paper is concisely written and has page limits as an
   extended abstract. But the readers of the LoG
   conferences are supposed to know GNNs and, ideally, they should not be
   assumed to know motion planning and reinforcement learning. Thus a
   potential issue would be that the paper might become inaccessible to
   these readers. For example,
   - At Line 38, the reader might have no idea of what the motion
     planning problem is.
   - At Line 130, the reader might have no idea of what the
     $\epsilon$-greedy strategy and temporal-difference learning are.
   In view of these, I would recommend the paper to include one or two
   sentences to explain these terminologies (at a high level), and
   give some references for further details.
4. As a matter of personal taste, I would like to have an exact
   definition and mathematical description of the proposed method. I
   believe that would improve the clarity. For example, at Lines
   84-86, there are three conditions for terminating the current
   episode, they could be translated into equivalent *if* statements.
   There are a few other such conditions at other places in the paper,
   and it would be nice to have an algorithmic listing that puts them
   together (in the appendix, perhaps, due to the limits of the space).


Minor: Typo at Line 80 (**a a reward**)

# Recommendation: Weak Accept (6)
I find the paper well written with better performance over existing
methods. I do list a few weaknesses and questions, but I think the strength outweighs them. This is a justification of my
recommendation for weak accept.

---

### Official Review · Reviewer_T9ze · 2022-10-22

**Overall Score:** 6
**Confidence:** 3

**Review:**

Contributions:
1. The paper focuses on an interesting problem of autonomous robotic assembly discovery (RAD). The goal is that a robotic agent should reason about abstract 3D target shapes that need to be fulfilled given a set of available building block.

2. The problem is challenging to solve .

3. The authors use a mixed-integer program formulation for global optimization of blocktype selection and use the blocks’ final position in order to recreate shapes.

4. The solution of the MILP’s is used as a guiding exploration signal in a graph Reinforcement Learning (RL) framework.

5. Authors use a GNN for capturing the geometric, structural, and physical relationship between entities.

6. They assume that all building blocks are a combination of primitive blocks. This helps with a modular representation of complicated blocks through primitive elements

 7. The authors train the GNN through model-free Q-learning. Further, they also show that it can be integrated with tree search(MCTS) for improved long-term decisions.

8. Empirically the results are significantly better than other methods and also generalize to out-of-distribution data.


Pros:
1. Good empirical evaluation.
2. Comparison with baseline
3. Ablation study to show importance of each component


Cons:
1. Presentation could have been better. For example, in sec.3 the dimensions of variables g, p,c should be clarified. Further, can you clarify how g becomes a vector after summation after grid state change in section 3. and how values remain <=1 in vector g.

2. Slight more explanation of how the scene is converted to a graph could be better. Although the authors have cited the paper. However, it will be better if some more details of  construction can be presented in the paper itself.

3. Code is not shared

Recommendation:
I vote for weak accept. There are some presentation issues that should be clarified. Further, the code is not shared.

---

### Meta-Review · Area_Chair_yHXr · 2022-11-14

**Confidence:** 4
**Recommendation:** Accept

**Meta Review:**

# Summary

This paper proposes to use a hybrid approach, based on reinforcement learning, graph neural network, and mixed-integer programming, to solve a 3D robot assembly discovery problem.
Briefly, the goal is to combine a predefined set of objects to form a new structure while considering task execution with the robot. Experiments are carried out on a set of  simulated and real-world environments and report both performance and robustness gains compared to an unstructured graph-based end-to-end approach.

#  Recommendation

This paper was well received by the reviewing team. We have a broad agreement towards an acceptance and the author response was helpful to converge towards this agreement. The concerns and questions were successfully addressed by the authors. Here are the three main points motivating this recommendation:

1. The problem discussed is interesting to address, and relates to a real-world application. As a bonus, it is nice to see videos showcasing the agent.

2. The two experiments are well-designed and propose good ablation studies. The first experiment illustrates that solving a MILP could yield a graph representation that is able to better guide the exploration of the GNNs + DQN. The second experiment shows that the reinforcement learning approach, which learns the action sequence as heuristics, is better than the heuristic designed by simple rules.
However, as pointed out in the reviews, it would have been interesting to see experiments against the MILP alone.

3. Due to the page limit, the paper needs to pack many information and assumes that the readers are familiar with RL in combinatorial optimization, MILP, etc. Additional explanations have been successfully addressed in the rebuttal and were added in the appendices. Consequently, the content of the paper has been significantly increased during the rebuttal (11 pages with the appendices) and the paper may now have a better fit as a regular paper.  Important information (RL environment with the reward definition, training algorithm, etc.) is now proposed in the appendix but will be more useful in the main text.

# Conclusion

To conclude, the reviewing team thinks that the paper deserves to be accepted (no spotlight). Besides, we strongly encourage the authors to incorporate the suggestions raised in the final version of the paper.

---

### Decision · Program_Chairs · 2022-11-22

Accept (Poster)